# Unveiling the Synergistic Effects of Phosphorus Fertilization and Organic Amendments on Red Pepper Growth, Productivity and Physio-Biochemical Response under Saline Water Irrigation and Climate-Arid Stresses

**DOI:** 10.3390/plants13091209

**Published:** 2024-04-26

**Authors:** Hamza Bouras, Krishna Prasad Devkota, Achraf Mamassi, Aicha Loudari, Redouane Choukr-Allah, Moussa El-Jarroudi

**Affiliations:** 1Department of Crop Production, Protection and Biotechnology, Hassan II Institute of Agronomy and Veterinary Medicine, Rabat 10101, Morocco; bourashamza07@gmail.com; 2International Center for Agricultural Research in the Dry Areas (ICARDA), Rabat 10100, Morocco; k.devkota@cgiar.org; 3SPHERES Research Unit, Department of Environmental Sciences and Management, University of Liège, B-6700 Arlon, Belgium; achraf.mamassi@inrae.fr; 4Unité de Recherche en Science Action Développement—Activités Produits Territoires (UMR-SADAPT), INRAE, AgroParisTech, Université Paris-Saclay, 91120 Paris, France; 5Plant Stress Physiology Laboratory-AgroBioSciences, Mohammed VI Polytechnic University (UM6P), Ben Guerir 43150, Morocco; aicha.loudari@gmail.com (A.L.); redouane53@yahoo.fr (R.C.-A.)

**Keywords:** salinity, irrigation, phosphorus, fertilization, organic matter, red pepper, dryland, Morocco

## Abstract

In regions facing water scarcity and soil salinity, mitigating these abiotic stresses is paramount for sustaining crop production. This study aimed to unravel the synergistic effects of organic matter and phosphorus management in reducing the adverse effect of saline water for irrigation on red pepper (*Capsicum annuum* L.) production, fruit quality, plant physiology, and stress tolerance indicators. The study was carried out in the arid Tadla region of Morocco and involved two key experiments: (i) a field experiment during the 2019 growing season, where red pepper plants were subjected to varying phosphorus fertilizer rates (120, 140, and 170 kg of P_2_O_5_.ha^−1^) and saline water irrigation levels (0.7; 1.5; 3; and 5 dS.m^−1^); and (ii) a controlled pot experiment in 2021 for examining the interaction of saline water irrigation levels (EC values of 0.7, 2, 5, and 9 dS.m^−1^), phosphorus rates (30, 36, and 42 kg of P_2_O_5_.ha^−1^), and the amount of organic matter (4, 8, 12, and 16 t.ha^−1^). The field study highlighted that saline irrigation significantly affected red pepper yields and fruit size, although phosphorus fertilization helped enhance productivity. Additionally, biochemical markers of stress tolerance, such as proline and glycine betaine, along with stomatal conductance, were impacted by increasing salinity levels. The pot experiment showed that combining organic amendments and phosphorus improved soil properties and stimulated red pepper growth and root weight across all salinity levels. The integration of phosphorus fertilization and organic amendments proved instrumental for counteracting salinity-induced constraints on red pepper growth and yield. Nonetheless, caution is necessary as high salinity can still negatively impact red pepper productivity, necessitating the establishment of an irrigation water salinity threshold, set at 5 dS.m^−1^.

## 1. Introduction

Salinity is a significant challenge that detrimentally affects agricultural productivity globally. Salinity is a significant challenge that detrimentally affects agricultural productivity globally. It is estimated that 20% of the world’s cultivated land and 33% of its irrigated agricultural areas are experiencing degradation due to salinity [1]. In Morocco, salinity and drought are the leading abiotic stresses impacting crop production, particularly in the irrigated regions. The Tadla Plain, a key agricultural zone covering 259,600 hectares, is facing substantial salinity issues, with 49% of its land used for irrigation [2]. This region’s arid climate and saline water irrigation are worsening soil salinity problems [3,4]. Notably, Tadla is the primary production area for red peppers in Morocco, fulfilling over 85% of the national demand [5]. However, soil salinity significantly restricts red pepper yields in the area [6,7]. Red pepper plants, which are moderately sensitive to salinity, have a tolerance threshold of 1.5 dS.m^−1^ [8]. Salinity levels exceeding this threshold can decrease marketable yields by up to 32%, and doubling the salinity to 3 dS.m^−1^ can lead to a 53% yield reduction [9]. Such declines are economically considered a significant loss for farmers. At higher salinity levels, red pepper growth is affected by complex physiological responses, including stomatal closure triggered by high salt concentrations in leaf apoplast, disruptions in photosynthesis partly due to lower chlorophyll content, and changes in carbon assimilation and utilization [10].

Phosphorus (P) is essential in plant physiology, playing a critical role in key metabolic processes such as photosynthesis, respiration, energy transfer, macromolecule biosynthesis, and signal transduction [11]. However, salinity and drought stress can significantly reduce phosphorus uptake [12], though phosphorus fertilization has been shown to potentially improve crop growth and productivity under these stresses [13]. The relationship between phosphorus availability and salinity stress is influenced by various factors including plant species, developmental stage, severity of stress, and environmental conditions like temperature, soil moisture, soil pH, and existing soil phosphorus levels [14].

Additionally, the use of soil organic amendments offers a sustainable solution by enriching soil fertility over the long term, increasing the availability of plant-accessible nutrients [15,16]. It is crucial to understand the combined effects of phosphorus fertilization and organic amendments to develop new strategies for improving the growth and physiology of red peppers, particularly in managing challenges posed by saline water irrigation and arid conditions. Prior studies have shown that integrating organic amendments with chemical fertilizers can lead to sustainable improvements in soil fertility and productivity, potentially increasing abiotic stress resilience in vegetable crops [17]. Therefore, exploring the interactions between irrigation water salinity, phosphorus fertilization, and the management of organic amendments, and their impact on red pepper cultivation, remains a promising area of research, especially in salt-affected and arid regions like the Tadla Plain. Thus, the major objective of this study was to explore how phosphorus fertilization and organic matter management influence red pepper growth, productivity, physiological development, and indicators of plant stress tolerance under saline water irrigation in arid climate conditions. Our research determines the complex interactions between various organic amendment and phosphorus fertilization strategies, aiming to highlight their significant effects on red pepper (*Capsicum annuum* L.) production, fruit quality, plant physiology, and key biochemical markers indicative of plant stress tolerance.

## 2. Materials and Methods

### 2.1. Soil and Climate Conditions

The experiments were conducted at the experimental station of the National Institute of Agronomic Research (INRA), Tadla, Morocco (Latitude = 32.2° N; Longitude = −6.31° W Altitude = 450 m). The field experiment was conducted between March to December 2019, while the pot experiment was conducted between March to June 2021. The soil of the experimental site was classified as Luvisols Chromic [18]. The climate in this area is arid with a remarkable irregularity of rainfall amount and distribution. Total annual rainfall is about 286 mm, and the average temperature is 18 °C, with a maximum temperature in August exceeding 45 °C and a minimum in January of approximately −3 °C (Figure 1).

Before starting experiments, following the protocol described by Jackson [19], soil analysis at two soil depths was performed (Table 1). The EC of the soil was measured using the soil-saturated paste method with an EC meter (HI 9812. Hanna Instruments, Smithfield, RI, USA). The recorded EC values were relatively low (Table 1), indicating that the soil can be considered non-saline [20]. Irrigation freshwater analysis is presented in Table 2 regarding EC, pH, cation, and anion concentrations.

### 2.2. Experimental Design, Treatments and Crop Management Practices

#### 2.2.1. Field Experimentation

A field experiment, applying four levels of irrigation water salinity and three P-rates, was conducted in a split-plot design with three replications (Figure 2) and using a Moroccan variety of red pepper “Lukkus”. The three P-rates evaluated were 120 (P1), 140 (P2), and 170 (P3) kg P_2_O_5_ kg ha^−1^. P1 is the recommended rate according to the soil status and crop species, P2 is the recommended rate of +20%, and P3 is the recommended rate + 40%. Four water salinity levels were evaluated: freshwater with electrical conductivity (ECw = 0.7 dS.m^−1^) and three saline water with EC of 1.5, 3, and 5 dS.m^–1^, respectively. The crop was sown on 8 April 2019 manually at a depth of 3 to 4 cm after the plowing and gentle compaction of the soil (to avoid soil being blown away by the wind). The area of each plot was 20 m^2^ (4 × 5 m^2^), composed of six rows with 70 cm row spacing (Figure 2). The nutrients were applied using fertigation with a drip irrigation system. Irrigation with saline water was started 30 days after planting, and the crop was irrigated daily up to the end of the experiment on 24 December 2019. The daily amount was modified each week according to the evapotranspiration (ET0) rate at the experimental station. The total irrigation water applied during the growing season was 606 mm. All plots with different salinity treatments received the same quantity of water in each irrigation event to avoid the confounding effect due to variation in irrigation amount.

Fruits were sampled four times during the growing cycle, once each month in August, October, November, and December. Fruits were harvested at the ripening (when it turns red) stage. At each sample, 40 fruits from each treatment were selected randomly to measure the quality parameters such as vertical diameter and fruit weight. The number of fruits per plant and the weight of each fruit were measured to determine the mean fruit weight and total fruit yield. Monthly ripe fruit harvests started 20 weeks after planting and continued for 125 days. Equatorial diameter and longitudinal diameter were measured using a digital vernier caliper.

#### 2.2.2. Pot Experimentation

The pot experiment was carried out under the greenhouse of the INRA experimental station in Tadla between March and June 2021. The same variety of red pepper (Lukkus) as the field experimentation was planted. The red pepper plants were transplanted on 8 March 2021 and harvested on 30 June 2021. The seedlings for the pot experiment were transplanted in plastic pots, each containing 15 kg of soil in three replications (Figure 3). The experiment was laid out in a split-plot design with two factors: (i) the water salinity (freshwater (0.7 dS.m^−1^) and three salinity levels of saline water irrigation: 2, 5, and 9 dS.m^−1^); and (ii) P-fertilization rate (three P rates). P fertilizer rates were applied at the seedling, which is commonly used in the region, in the form of triple superphosphate (45% of P_2_O_5_) at 30, 36 (+20%), and 42 (+40%) kg P_2_O_5_ ha^−1^. Organic amendments, including compost and an industrial amendment with a substantial 30% organic matter content, were meticulously applied across four distinct levels within the experimental sub-plots.

Organic matter was applied at the rates of 4, 8, 12, and 16 t.ha^−1^ to the seedlings, mixed directly with the soil. At the seedling stage, nitrogen fertilizer was applied uniformly as ammonium sulfate (21% N) at a rate of 1.4 g N per pot. Both fertilizers and organic amendments were mixed thoroughly and transferred to plastic pots before seedling transplanting. After transplanting, the pesticide Imidacloprid was applied against aphids. For the first ten days, seedlings were irrigated with fresh water at 0.7 dS.m^−1^. Additional nitrogen requirements were applied equally for all treatments through fertigation with a drip irrigation system. Saline water irrigation started 15 days after planting and continued two to three times until the harvest on 30th June 2021, following the ET0 method.

### 2.3. Measurements

#### 2.3.1. Growth Parameters and Fruit Yield

Sixty days post-seedling, three plants from each treatment were randomly collected for the analysis of growth parameters. The assessed parameters included the leaf area, number of leaves per plant, stem and main root length, dry weight of shoot and roots, and leaf content. The shoot and root lengths of three randomly selected plants were measured from every pot, and the average values were calculated. The fresh weight of the shoot and root was determined separately with the help of a digital electrical balance. Then, samples were dried in an oven at 60 °C for 48 h, and the dry weight was measured. In the field experiments, the fruit yield was quantified at harvest by assessing the total production from the entire plot area of 20 m^2^.

#### 2.3.2. Physiological Parameter (Stomatal Conductance)

Stomatal conductance was determined using the SC-1 Leaf Porometer (Decagon Devices Inc., Pullman, WA, USA). By quantifying vapor flux from the leaf through the stomata, this device facilitates the estimation of the variance between transpiring leaves and those that have ceased transpiration. It was measured midday between 11 AM and 3PM on the upper leaf surface well exposed to sunlight. Each plant was measured once every two months, with six plants evaluated per plot.

#### 2.3.3. Biochemical Parameters (Proline and Glycine Betaine)

Proline content was determined spectrophotometrically from leaves harvested at 45, 60, and 75 days after transplanting, following the modified ninhydrin method described by [21] and using (L proline) as a standard. The glycine betaine (GB) content of red pepper was quantified using the procedure described previously by [22]. The concentration of GB was measured using the standard curve, and the results were expressed as μg.g^–1^ of dry matter.

#### 2.3.4. Soil Salinity

In situ EC was measured using a conductivity meter that allowed us to measure soil EC in different soil layers and depths. Soil salinity was measured in December 2019 (after the last saline water irrigation dose) in the field experimentation. In the pot experiment, EC was measured every month from each plot and treated at 0–10 to 10–20 cm pot depths (from March to June 2021).

### 2.4. Statistical Analysis

Data were analyzed using the analysis of variance (ANOVA) applying ‘agricolae’ package of R Version 4.0.4. Before conducting the ANOVA, the normality of the yield distribution for each factor (salinity, P, and organic amendments) was examined using the Shapiro–Wilk test. A multifactor ANOVA was carried out to define the significant factors (irrigation water salinity, P rate, and organic matter) and their interactions. When the ANOVA test was significant, we performed Tukey’s test (*p* = 0.05) to identify the significant differences between treatments.

## 3. Results

### 3.1. Red Pepper in Field Experimentation

#### 3.1.1. Salinity × P-Fertilization Interaction Effects on Stomatal Conductance

Saline water irrigation has significantly reduced red pepper’s stomatal conductance (gs) (Figure 4A). The S1 (EC 0. 7 dS.m^−1^) treatment resulted in the highest opening of the leaf stomata and, consequently, better transpiration by the plant (Figure 4). The negative effect of salinity on stomatal conductance became noticeable from the electrical conductivity of 1.5 dS.m^−1^. Compared to the control, irrigation water salinity decreased the stomatal conductance by 28%, 41%, and 53%, respectively, under S2 (1.5 dS.m^−1^), S3 (3 dS.m^−1^), and S4 (EC 5 dS.m^−1^) (Figure 4B). Additionally, a higher phosphorus application rate of 170 kg.ha^−1^ P_2_O_5_ notably enhanced stomatal conductance, particularly under salinity levels of 3 and 5 dS.m^−1^.

#### 3.1.2. Salinity × P-Fertilization Interaction Effects on Fruit Yield of Red Pepper

Salinity significantly impacted the fruit yield of red pepper, with reductions starting at a salinity level of 1.5 dS.m^−1^ and reaching the highest decrease at 5 dS.m^−1^ (Figure 5A). The yield declined at a rate of 1.26 tons per hectare, exhibiting a strong linear negative correlation with a coefficient of determination (R²) of 0.99. Relative to the control, yields under saline conditions decreased by 12%, 16%, and 28% at salinity levels of 1.5, 3, and 5 dS.m^−1^, respectively.

Regarding the interaction between P-fertilizer and irrigation water salinity, P-application increased fruit yield even under saline conditions (Figure 5B). With freshwater application, P-application increased yield by 5%. In saline environments, the additional P-fertilizer supply significantly increased the yield of red pepper, where P2 (140 kg. ha^−1^ of P_2_O_5_) boosted yield by 11% at an EC level of 1.5 dS.m^−1^. Under a salinity level of 3 dS.m^−1^, the additional P-application increased (*p* < 0.05) yield by 4% and 9%, respectively, with P2 and P3 (170 kg. ha^−1^ of P_2_O_5_), as compared with the control (P1 = 120 kg. ha^−1^ of P_2_O_5_). At the highest salinity level (EC 5 dS.m^−1^), red pepper yield improved by 13% under P2 and 17% under P3, as compared with the control. The analysis of variance confirmed that the interaction between salinity and phosphorus significantly affected fruit yield (*p* < 0.05).

#### 3.1.3. Salinity × P-Fertilization Interaction Effects on Fruit Diameter of Red Pepper

The fruit diameter of red pepper was significantly affected by salt treatments (Table 3). Similar to the tendency in fruit yield, salinity significantly reduced red pepper fruit size. The vertical diameter of fruit was reduced by 6% and 12%, respectively, under 3 and 5 dS.m^−1^ as compared with the control. However, no significant effect was recorded on equatorial diameter with saline water (EC = 1.5 dS.m^−1^) and fresh water. Similar results were obtained for the longitudinal diameter of the fruit. The longitudinal diameter was significantly decreased by saline irrigation by 5%, 9%, and 10%, under 1.5, 3, and 5 dS.m^−1^, respectively, compared with the control (Table 3). Fruit weight was also significantly reduced by salinity stress, decreasing by 26%, 30%, and 25% under salinity levels of 1.5, 3, and 5 dS.m^−1^, respectively, compared to the control.

Phosphorus application significantly enhanced the fruit’s weight and longitudinal and vertical diameter under salt stress conditions. The beneficial effect of P-fertilization appears after an EC level of 1.5 dS.m^−1^ and persists up to 5 dS.m^−1^ (Table 3). At an EC of 1.5 dS.m^−1^, the phosphorus dose (P2) resulted in a 4% increase in fruit weight compared to the lower dose (P1). At a moderate salinity level (EC 3 dS.m^−1^), P2 notably increased fruit weight by 6% relative to P1. Under the highest salinity, fruit weight increased by 6% and 10% for P2 and P3, respectively, compared to P1.

Phosphorus supply also improved the longitudinal diameter of the fruit. The longitudinal diameter varied slightly with increasing irrigation water salinity up to 5 dS.m^−1^. However, no significant effect was noticed at EC below 1.5 dS.m^−1^. P2 exhibited the most substantial increases in longitudinal diameter under EC levels of 3 and 5 dS.m^−1^, with increases of 14% and 6%, respectively, compared to EC 1.5 dS.m^−1^. Similarly, phosphorus application enhanced the vertical diameter of the fruits; P2 significantly increased the vertical diameter by 9% at EC 3 dS.m^−1^, while P3 led to a 4% increase at EC 1.5 dS.m^−1^ compared to P1.

### 3.2. Interaction Effects of Organic Matter and P-Fertilization with Saline Water Irrigation on Soil Salinity, and Red Pepper under Pot Experiment

#### 3.2.1. Salinity × (Organic Matter-P Supply) Interaction Effects on Soil Salinity

Soil salinity varied with different phosphorus rates and organic matter applications (Figure 6). Saline water irrigation tends to accumulate more salt at a depth of 0–10 cm than at 10–20 cm. Figure 6 indicates that under freshwater conditions, soil salinity was unaffected by varying rates of organic matter and phosphorus. However, at a water salinity of 2 dS.m^−1^, a P-rate of 36 kg ha^−1^ resulted in lower soil salinity, whereas the same P-rate led to the highest salt accumulation in the soil at a water salinity of 5 dS.m^−1^. Excluding freshwater, an organic matter dose of 4 t ha^−1^ was associated with higher salinity accumulation compared to other levels of organic matter (Table 4).

#### 3.2.2. Salinity × (Organic Matter-P Supply) Interaction Effects on Proline Accumulation

Salinity stress notably increased proline accumulation, indicative of plant stress, with significant increases observed under the highest salinity levels of 5 and 9 dS.m^−1^ (Figure 7). At lower salinity levels of 0.7 and 2 dS.m^−1^, neither phosphorus supply nor organic matter had a significant effect on leaf proline accumulation. However, at 5 dS.m^−1^, proline accumulation was notably higher at the lowest phosphorus rate (P1 = 30 kg ha^−1^). Phosphorus supply helped mitigate the stress effects (resulting in lower proline accumulation) at this salinity level. In contrast, at the higher salinity level of 9 dS.m^−1^, the addition of organic amendments led to increased proline accumulation in red pepper leaves, whereas increasing the phosphorus rate did not significantly affect proline accumulation.

#### 3.2.3. Salinity × (Organic Matter-P Supply) Interaction Effects on Glycine Betaine Accumulation

Salinity stress has led to increased GB accumulation, indicative of plant stress, in the leaves of red pepper (Figure 8). While there was no significant difference in GB levels among salinity levels of 0.7, 2, and 5 dS.m^−1^, plants tended to accumulate more GB at a higher salinity level of 9 dS.m^−1^. At salinity levels of 0.7 and 2 dS.m^−1^, a phosphorus supply rate of 36 kg ha^−1^ significantly increased GB accumulation. However, at higher water salinity levels of 5 and 9 dS.m^−1^, increased phosphorus supply was associated with reduced GB levels in the leaves. The study also found that higher rates of organic amendments (12 and 16 tons ha^−1^) significantly led to lower GB accumulation. This suggests that increasing organic matter in the soil tends to reduce GB accumulation, thereby potentially mitigating plant stress under saline conditions.

Phosphorus supply at a rate of 36 kg ha^−1^ significantly increased GB accumulation, particularly at lower salinity levels of 0.7 and 2 dS.m^−1^. However, at moderate (5 dS.m^−1^) and high salinity levels (9 dS.m^−1^), P-supply led to a reduction in GB content in the red pepper leaves. Concerning organic amendments, increasing the amount of organic matter resulted in decreased GB accumulation, as depicted in Figure 5.

#### 3.2.4. Salinity × (Organic Matter-P Supply) Interaction Effects on Stomatal Conductance

Increased levels of saline water irrigation significantly decreased the stomatal conductance in red pepper (Figure 9). Regarding phosphorus supply, higher P rates significantly enhanced stomatal conductance; the rate of 42 kg ha^−1^ resulted in the maximum stomatal conductance. Across all salinity levels, the lowest stomatal conductance occurred with the highest application of organic amendments at 16 t. ha^−1^.

#### 3.2.5. Salinity × (Organic Matter-P Supply) Interaction Effects on Root Development

Results indicate significant variations in the dry weight of red pepper roots per plant across different salinity levels (Figure 10). As water salinity increased, there was a notable decrease in root dry weight by 33%, 54%, and 77%, respectively, at salinity levels of 2, 5, and 9 dS.m^−1^, compared to the control (0.7 dS.m^−1^). Figure 10 also shows that increasing P-fertilization rates significantly improved root development, with the highest root weight achieved at a dose of 42 kg ha^−1^. Similarly, the application of organic amendments demonstrated a beneficial effect, where a high supply of 16 tons/ha resulted in the greatest root weight across all water salinity levels.

## 4. Discussion

### 4.1. Salinity × P-Fertilization Interaction Effects on Red Pepper Stomatal Conductance

Results related to stomatal conductance demonstrated that salinity levels from irrigation water negatively impact this physiological parameter (Figure 4 and Figure 9). Stomatal conductance is regulated primarily by the aperture of the stomatal pore and stomatal density, as well as the water transport capacity of the guard cells on the leaf surface [19]. An increase in salinity levels significantly affected red pepper stomatal opening rate, following a linear negative relationship with R^2^ equal to 0.95. This finding is consistent with previous research on the effects of salt stress on plants [23,24], which indicates that salt stress causes osmotic stress in the roots, leading to decreased water uptake and the subsequent closure of stomata to conserve water within the plant tissues and reduce water loss through transpiration. As a compensatory mechanism, plants adjust their net CO_2_ assimilation and transpiration rates [25]. However, severe salt stress can lead to the dehydration of mesophyll cells, disrupting photosynthetic activity and reducing root hydraulic conductivity [26]. Another study [27] noted a decrease in leaf water potential due to stomatal closure, even though turgor was maintained in mature leaves when red pepper was irrigated with saline water. As for the impact of P-supply on stomatal conductance, it significantly enhanced transpiration in the leaves of red peppers under saline conditions (Figure 4B), while showing no significant effect under freshwater conditions.

### 4.2. Salinity and P-Fertilization Interaction Effects Yield Parameter of Red Pepper

Salinity stress inhibits the process of photosynthesis by perturbing the biochemical mechanism of CO_2_ fixation or the limitations in the CO_2_ supply arising from stomatal closure [10,11,12,13,14,15,16,17,18,19,20,21,22,23,24,25,26,27,28]. Our results indicated that red pepper’s fruit yield significantly declined under salinity stress (Figure 5), aligning with studies by [29,30], which noted severe reductions in red pepper fruit production due to salinity. This reduction in total yield is attributed to smaller fruit size and the decreased number and weight of fruits [9]. The drop in fruit yield can be traced back to the decreased leaf stomatal opening (Figure 4 and Figure 9), which lowers photosynthesis rates. Additionally, it impaired the plant’s water uptake and growth rate, thereby reducing yield alongside various metabolic changes similar to those caused by water stress [31]. Moreover, salt stress reduces macro-nutrient uptake in pepper plants [6], which diminishes the energy available for nutrient translocation, leading to increased sodium ion (Na^+^) accumulation that depolarizes cellular membranes [32], and creates antagonism between Na^+^ and other vital minerals like K^+^, Ca^2+^, Mg^2+^, and P^3+^ [33]. An increase in salinity also affected the number and average weight of fruits (Table 3), suggesting that salinity might lead to flower or fruit abortion and reduced fruit weight. However, contrary findings by [34] observed that at the same salinity levels, the number of fruits, both total and marketable, remained largely unaffected or only slightly impacted, indicating that the reduction in average fruit weight was the primary factor behind yield reduction. A distinct reduction in the fruit yield of red pepper from a salinity level of 1.5 dS.m^−1^ (Figure 5) aligns with the findings by [6], which recorded a similar impact on all the growth parameters of pepper plants beyond this salinity threshold.

Application of P-treatments, particularly P3 (170 kg ha^−1^), caused an increment in fruit yield under saline conditions. This improvement is likely due to phosphorus’s role in enhancing numerous metabolic processes including energy transfer, signal transduction, the biosynthesis of macromolecules, photosynthesis, and respiration [11]. These results highlight findings from previous studies, like [35], which demonstrated that phosphorus application could partially counteract the adverse effects of salinity on maize. Overall, the beneficial effects of phosphorus fertilization on crop yield were evident across diverse salinity levels, showcasing significant enhancements for mung bean [36] and chickpeas [37], while demonstrating somewhat lesser impacts on other crops, such as wheat [38,39].

### 4.3. Salinity × (Organic Matter-P Supply) Interaction Effects on Red Pepper Biochemical Traits

Glycine betaine (GB) and proline are two majors organic osmolytes that accumulate in various plant species in response to environmental stresses such as drought, cold, and salinity [40]. Our findings show that salinity led to increased proline accumulation in the leaves of red pepper plants (Figure 7). The variations in proline levels were influenced by changes in both irrigation water salinity and phosphorus fertilization (Figure 4). Previous studies have shown that proline production is enhanced under salinity stress in plants such as lentils [40], rice [41], canola [42], and wheat [43]. Proline helps regulate genes related to antioxidant enzymes under salt stress, alleviating the negative impact of salt by reducing osmotic stress, thus maintaining normal membrane function and integrity. Its accumulation in plants offers protection against extreme salinity and drought stresses [44] and is linked to cellular osmotic adjustment, detoxifying excess ammonia, stabilizing proteins and/or membranes, and enhancing the stability of certain cytoplasmic and mitochondrial enzymes [43].

Similarly, salinity induced an accumulation of GB in the leaves of red pepper plants (Figure 8). Previous studies have reported that GB accumulates in response to stress in many crops, including barley and wheat [45]. Glycine betaine has been found to significantly enhance salt tolerance in common beans by boosting antioxidant defenses, both enzymatic (e.g., peroxidase, superoxide dismutase, catalase) and non-enzymatic (e.g., proline, glutathione) [46]. The biosynthesis of GB is stress-inducible [47], and its level of accumulation varies with the plant’s salt tolerance [48]. Glycine betaine plays a crucial role in maintaining osmotic regulation in cells [49], is primarily found in the chloroplast, and is vital in protecting the thylakoid membrane, thus maintaining photosynthetic efficiency. Glycine betaine also supports several transporters in functioning effectively under salt stress, protecting plants by promoting a high K^+^/Na^+^ ratio and enhancing antioxidant defenses by limiting Na^+^ uptake [46].

### 4.4. Salinity × (Organic Matter-P Supply) Interaction Effects on Soil and Root Development

Stomatal conductance is crucial for maintaining water balance, and its reduction under moderate salt stress primarily affects plant growth by limiting cell expansion and development, thereby reducing biomass and productivity [50]. Our study showed that applying organic amendments and phosphorus can enhance the physical and chemical properties of saline soils and promote red pepper growth [51]. This finding aligns with research that reported green waste compost improving soil properties and tree growth in saline coastal areas of northern China [16].

Our results also demonstrated that phosphorus supply was more effective, significantly increasing root weight in red pepper compared to organic amendments (Figure 10). The highest phosphorus rate led to the maximum root weight; a finding similarly observed with high organic amendment levels under all tested water salinity levels. Salinity stress impacts include water stress, ion toxicities, and ion imbalance [52]. Overall, applying phosphorus fertilizer at all salinity levels improved both above- and underground plant parts. Previous studies have shown that higher phosphorus rates can mitigate the decline in plant height in saline soil [52,53], with positive impacts on shoot and root biomass, length, phosphorus concentration, and chlorophyll content in green beans, as well as increasing root length and weight in maize and sugar beet under saline conditions [4,5,6,7,8,9,10,11,12,13,14,15,16,17,18,19,20,21,22,23,24,25,26,27,28,29,30,31,32,33,34,35,36,37,38,39,40,41,42,43,44,45,46,47,48,49,50,51,52,53,54].

## 5. Conclusions

Salinity has significantly reduced red pepper’s yield and physiological development parameters. It was also observed that plants grown in saline water treatments 0.7 (control) and 1.5 dS.m^−1^ were similar in response to salinity. The findings of the current study suggest that despite the positive effects of P-fertilization, irrigation with saline water has an EC exceeding 5 dS.m^−1^ significantly and irreversibly affects the growth and yield of red pepper. Hence, it is strongly advised against irrigating red pepper cropping systems with saline water having an EC over 5 dS.m^−1^. Supplementary P-supply, with P2 (140 kg P_2_O_5_ ha^−1^), is recommended for increasing the yield of red pepper around EC of 1.5 dS.m^−1^, and P3 (170 kg P_2_O_5_ ha^−1^) for EC exceeding 3 dS.m^−1^. Similarly, in the pot experiments, the proline, glycine betaine (GB), and stomatal conductance were correlated negatively with the increasing salinity, especially under high levels of saline water (5 and 9 dS.m^−1^). However, the adverse effects of salinity could be partially alleviated through P-fertilization and organic matter management. The results from the pot experiment highlight that applying an optimal rate of phosphorus and organic fertilizers (such as compost or industrial amendments with 30% organic matter) to saline soil could serve as a strategy to mitigate the adverse effects of salinity stress and enhance growth. Our research underscores the critical importance of effectively managing saline water irrigation and employing tailored fertilization techniques to counteract the adverse impacts of salinity stress on red pepper cultivation. Through the implementation of precise phosphorus fertilization methods and the integration of organic amendments into soil management practices, farmers can bolster the growth and yield of red pepper crops in saline-affected regions. This proactive approach not only enhances agricultural productivity but also fosters the sustainability and resilience of farming systems.

## Figures and Tables

**Figure 1 plants-13-01209-f001:**
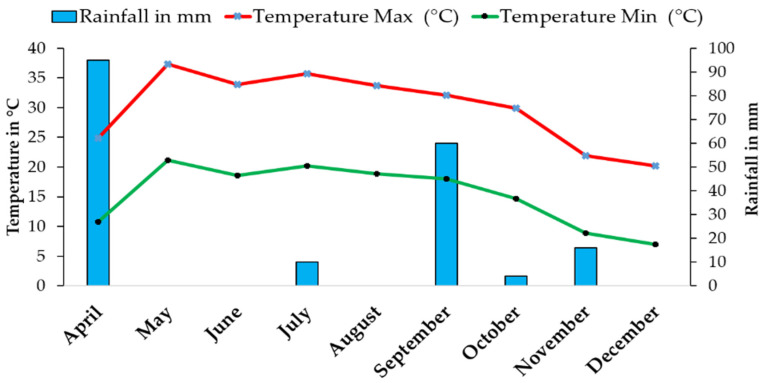
Temperature and rainfall variations at the experimental station throughout the 2019 crop-growing season.

**Figure 2 plants-13-01209-f002:**
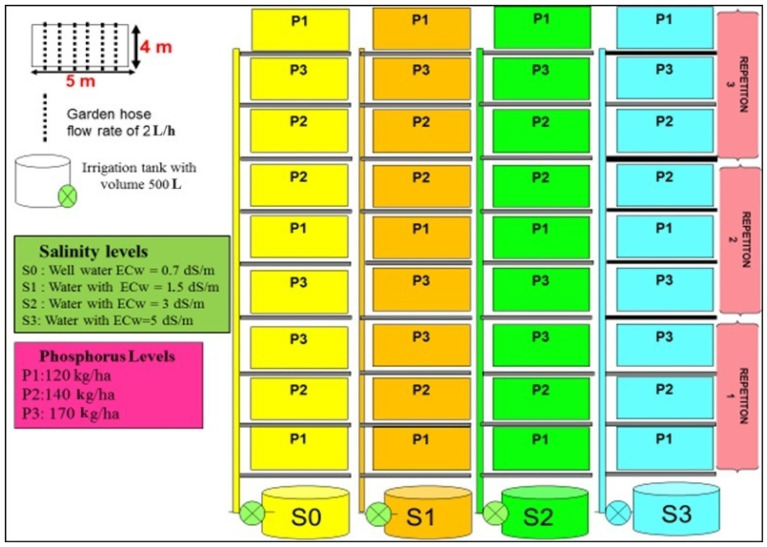
Design and arrangement of field experimental plots.

**Figure 3 plants-13-01209-f003:**
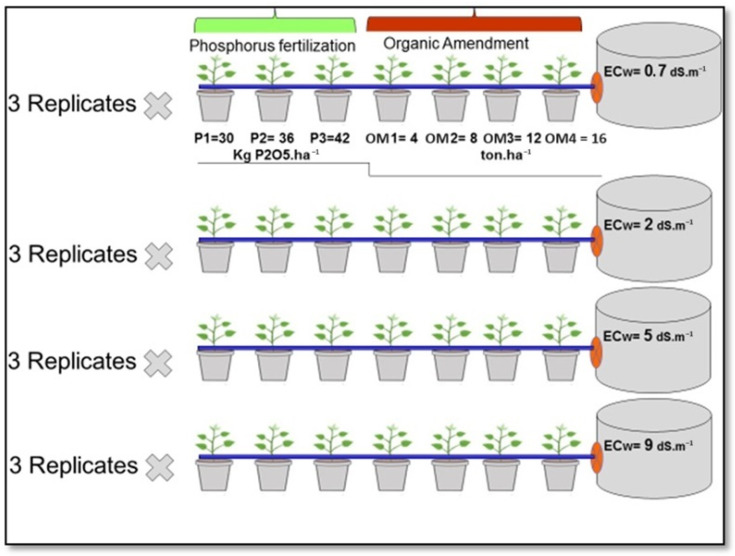
Experimental protocol for pot trials: investigating the effects of salinity and Organic Matter-P Fertilization treatments.

**Figure 4 plants-13-01209-f004:**
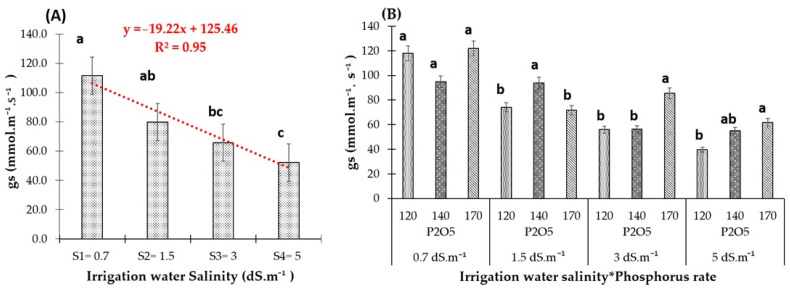
(**A**) Stomatal conductance of red pepper as affected by four irrigation water salinity levels. (**B**) Stomatal conductance of red pepper as affected by the interaction between the three phosphorus fertilizer rates and each of the four irrigation water salinity levels. Bars with the same letter are not significantly different according to the least significance difference test at *p* ≤ 0.05. Vertical lines are standard errors.

**Figure 5 plants-13-01209-f005:**
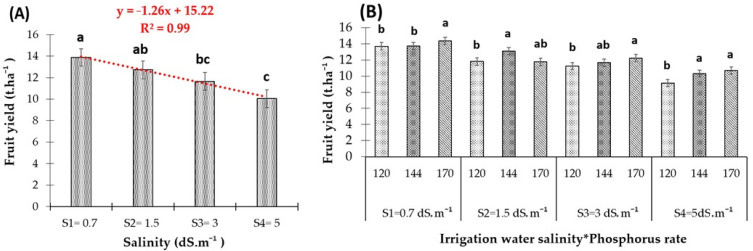
(**A**) Effect of irrigation water salinity on the fruit yield of red pepper. (**B**) Interactive effects of the three phosphorus fertilizer rates and each of the four irrigation water salinity levels on the fruit yield of red pepper. Bars with the same letter are not significantly different according to the least significance difference test at *p* ≤ 0.05. Vertical lines are standard errors.

**Figure 6 plants-13-01209-f006:**
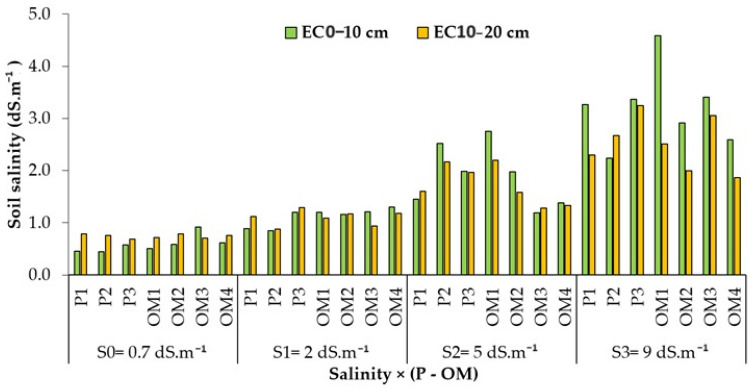
Variation of the soil salinity of the red pepper in pot experiment under different levels of irrigation water salinity.

**Figure 7 plants-13-01209-f007:**
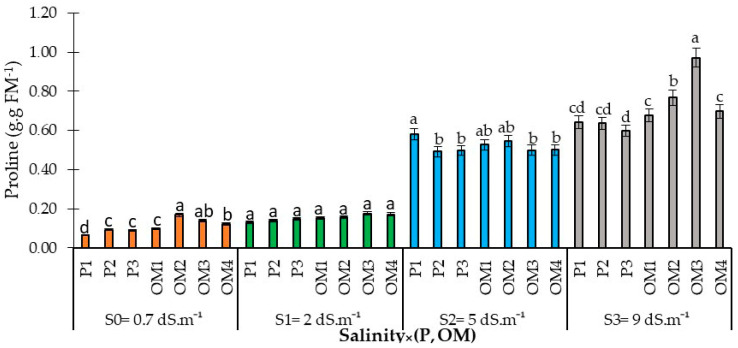
Proline content in red pepper as affected by saline water irrigation interaction with phosphorus and organic matter supply. Error bars indicate the standard deviation. Bars with the same letter are not significantly different according to the least significance difference test at *p* ≤ 0.05. Salinity × (P, OM): the dynamic interaction between salinity levels and two factors: Phosphorus fertilization and Organic Matter levels.

**Figure 8 plants-13-01209-f008:**
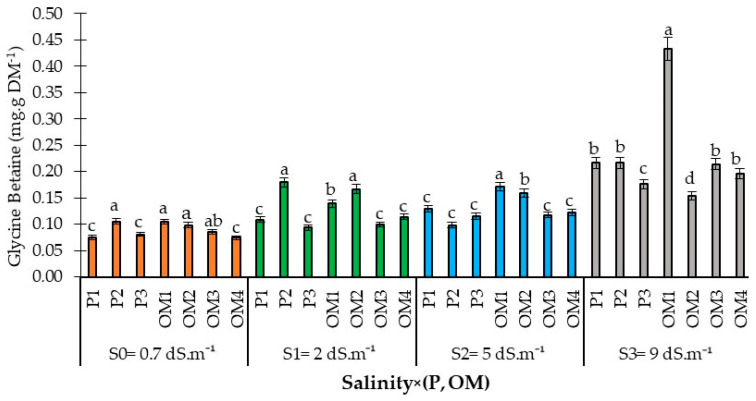
Glycine betaine content in red pepper leaves as affected by the interaction between saline water irrigation, phosphorus, and organic matter supply. Error bars represent the standard deviation. Bars with the same letter are not significantly different according to the least significance difference test at *p* ≤ 0.05. Salinity × (P, OM): the dynamic interaction between salinity levels and two factors: Phosphorus fertilization and Organic Matter levels.

**Figure 9 plants-13-01209-f009:**
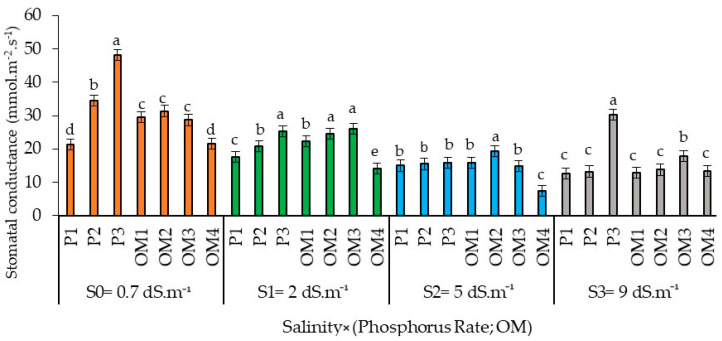
Stomatal conductance of red pepper as affected by saline water irrigation interaction with phosphorus and organic matter supply. Error bars indicate the standard deviation. Bars with the same letter are not significantly different according to the least significance difference test at *p* ≤ 0.05. Salinity × (Phosphorus rate; OM): the dynamic interaction between salinity levels and two factors: Phosphorus fertilization and Organic Matter levels.

**Figure 10 plants-13-01209-f010:**
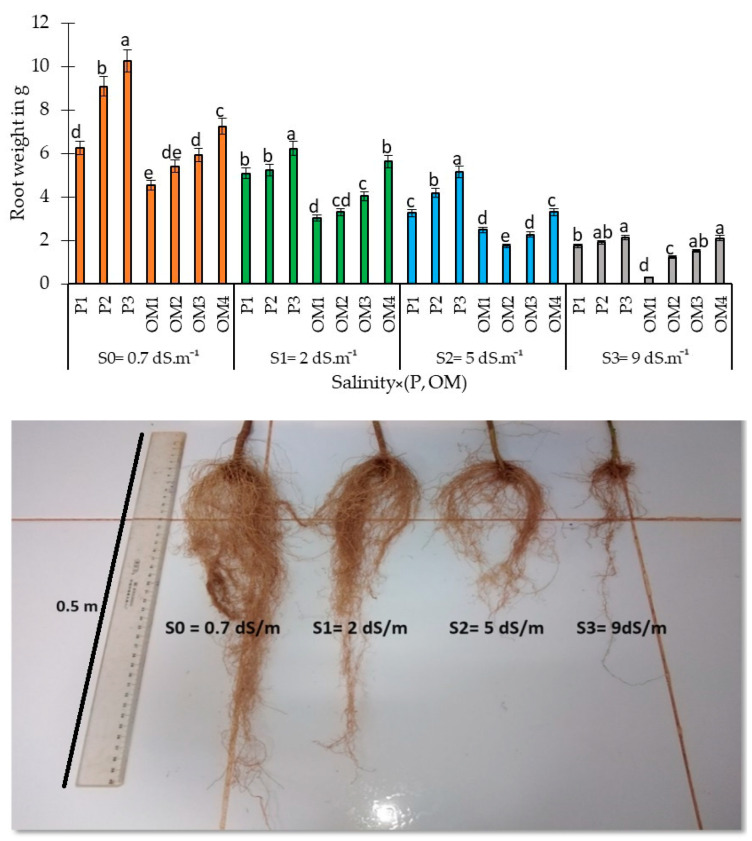
Variation of root development (weight and length) under different saline water irrigation levels. Bars with the same letter are not significantly different according to the least significance difference test at *p* ≤ 0.05. Salinity × (P, OM): the dynamic interaction between salinity levels and two factors: Phosphorus fertilization and Organic Matter levels.

**Table 1 plants-13-01209-t001:** Initial soil characteristics in the experimental site.

Soil Depth (cm)	Clay (%)	Silt (%)	Sand (%)	Soil pH	pH		EC (dS·m^−1^)	Organic Matter (%)	Total N (Kjeldahl) (g.kg^−1^)	P_2_O_5_ (Olsen) (mg.kg^−1^)	K_2_O (Acetate of Na) (mg.kg^−1^)
					Water	KCl					
0–20	28.1	52.8	19.1	7.92	8.24	7.36	0.1	1.45	2.34	43	459
20–40	43.1	18.7	38.2	8.09	8.38	7.24	0.22	0.59	3.44	22	405

**Table 2 plants-13-01209-t002:** Irrigation freshwater chemical analysis.

EC (dS.m^−1^)	pH	Cations (meq/L)				Anions (meq/L)				
		Ca^2+^	Mg^2+^	Na^+^	K^+^	Cl^−^	SO_4_^2−^	CO_3_^2−^	HCO_3_^−^	NO_3_^−^
0.7	7.4	2.4	3.9	2.29	0.001	2.25	0.54	1.2	4.3	0.124

**Table 3 plants-13-01209-t003:** Interaction effect of irrigation water salinity and phosphorus fertilizer rate on longitudinal diameter, equatorial diameter, and in the fruit weight of red pepper.

Salinity (dS.m^−1^)	Phosphorus Rate (kg P_2_O_5_.ha^−1^)	Fruit Weight (g)	Longitudinal Diameter (mm)	Equatorial Diameter (mm)
0.7	120	13 ± 2 a	30 ± 5 a	40 ± 3 a
	140	13 ± 2 a	27 ± 2 a	42 ± 3 a
	170	11 ± 2 a	28 ± 4 a	40 ± 3 a
	Average	12 ± 2 A	28 ± 4 A	41 ± 3 A
1.5	120	10 ± 3 a	26 ± 3 a	40 ± 5 ab
	140	7 ± 3 b	24 ± 3 a	38 ± 3 b
	170	7 ± 2 ab	25 ± 3 a	43 ± 3 a
	Average	8 ± 2 B	25 ± 3 B	40 ± 4 A
3	120	9 ± 2 a	25 ± 3 a	38 ± 2 a
	140	7 ± 1 a	23 ± 2 a	38 ± 2 a
	170	8 ± 2 a	23 ± 1 a	34 ± 4 b
	Average	8 ± 2 B	24 ± 2 B	37 ± 3 B
5	120	9 ± 2 a	24 ± 2 ab	37 ± 2 a
	140	10 ± 2 a	27 ± 2 a	38 ± 4 a
	170	9 ± 2 a	24 ± 2 b	32 ± 3 b
	Average	9 ± 2 B	25 ± 2 B	36 ± 3 B

Lowercase and same letters (a, ab, b) indicate the statistically homogeneous groups within phosphorus fertilization treatments, and uppercase and same letters (A, AB, B) indicate the statistically homogeneous groups within salinity.

**Table 4 plants-13-01209-t004:** Variation of soil salinity (EC) in different soil depths under different irrigation water salinity in the field (2019) and pot experiments (2021).

Depth in cm	Water Salinity (dS.m^−1^)							
	Field Trial (2019)				Pot Trial (2021)			
	0.7	1.5	3	5	0.7	2	5	9
0–10	0.8 ± 0 b	1.2 ± 0.1 b	1.9 ± 0 a	2.1 ± 0 a	0.6 ± 0.1 C	1.1 ± 0 C	1.9 ± 0.1 B	3.2 ± 0.1 A
10–20	0.9 ± 0.1 c	1.4 ± 0 b	3.1 ± 0.6 a	2.4 ± 0.1 ab	0.7 ± 0 C	1.1 ± 0 C	1.7 ± 0 B	2.5 ± 0.1 A
20–30	1 ± 0.1 c	2.6 ± 0.2 b	3.7 ± 0.3 a	3.5 ± 0.1 a	-	-	-	-
30–40	0.8 ± 0.1 d	1.7 ± 0.2 c	3.1 ± 0.2 b	4 ± 0.1 a	-	-	-	-

Means with the same letters do not differ significantly at *p* = 0.05.

## Data Availability

Data are contained within the article.

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
