# Peer review of "Unveiling the Synergistic Effects of Phosphorus Fertilization and Organic Amendments on Red Pepper Growth, Productivity and Physio-Biochemical Response under Saline Water Irrigation and Climate-Arid Stresses"

_plants, 2024, doi:10.3390/plants13091209_

Round 1

Reviewer 1 Report

Comments and Suggestions for Authors

The conducted research is valid, but only in one growing season (annual). It is not possible to draw conclusions based on one year of field research. The research must be at least two years old. According to the agrotechnical research methodology, they should be three years old. I believe that research should be continued and only after 2 or 3 years of research can proper conclusions be drawn.

Comments on the Quality of English Language

The English language requires minor changes

Reviewer 2 Report

Comments and Suggestions for Authors

The manuscript titled " Unveiling the Synergistic Effects of Phosphorus Fertilization and Organic Amendments in Mitigating Saline Water Irrigation and Dryland Stresses: Assessment of Red Pepper (Capsicum annuum L.) Growth, Productivity and Physio-Biochemical changes" submitted by Bouras et al. had investigated the synergistic effects of organic matter and phosphorus management in alleviating the impact of saline water irrigation on red pepper. This work has some certain application value. However, before this manuscript become acceptable, it needs further deep revision.

1, the title is too long and not concise enough.

2, Fig.1, 2, 3 are lacking clear and understandable figure legends.

3, Fig.10 and 11 should be merged into one plate. And fig.10 should be appropriately reduced in size.

4, The author did not have a good discussion on why only stomatal conductance is considered in photosynthetic indicators. In fact, Fv/Fm is a more convincing indicator for evaluating plant stress resistance. And I also hope that the author can provide other photosynthetic indicators such as net photosynthetic rate and transpiration rate.

Reviewer 3 Report

Comments and Suggestions for Authors

This paper  has certain reference significance for guiding the production and cultivation of red pepper under salt stress and the improvement and utilization of saline alkali land. But the original manuscript cannot be published in the journal in its current form. If the author can make supplements and modifications in the following aspects, it will be helpful for the acceptance of the manuscript.

1. The experiments were conducted using field experiments and pot experiments, but the years, locations, soil conditions, testing methods, and treatment levels of the two experiments were different. How to ensure the consistency and reliability of the test results should be explained in the discussion.

2. The experiment used three repeated split plot design, why did not the ANOVA results of the split plot design be listed? If this result is available, it can clearly show the main and interactive effects of phosphorus fertilizer treatment, irrigation water treatment, and organic matter treatment.

3. The significance difference test in Figures 4 (B) and 5 (B) should be conducted separately for four different irrigation water salinity treatments, which should be explained in the caption of the figure.

4. The representation of the four treatment levels of organic matter is shown in Figure 3 as OA1, OA2, OA3, and OA4 (written as OA in the figure), while Figures 6, 7, 8, 9, and 11 are OM1, OM2, OM3, and OM4, which should be consistent.

5. Line 266~268: P-application has a significant positive effect on red pepper fruit’s weight and longitudinal and vertical diameter under salt stress conditions. The positive effect of P fertilization appears after an EC level of 1.5 dS.m-1 and continues up to 5 dS.m-1 (Table 2), but Table 2 is Irrigation freshwater chemical analysis (line 117).

6. The title of Table 4 states that "values with the same letters are statistically equal at the same salinity level" and the annotation in Table 4 states that "there is no significant difference in the mean value with the same characters at p=0.05". The former should be deleted, and the statement “Error bars indicate the standard deviation” should be deleted.

7. In discussion 4.2, the author believes that Under salinity stress, the reduction of fruit yield can be explained by dropped leaf stomatal opening (Figure 4; Figure 9), which reduces photosynthetic rates (line 386-388). The author mentioned in section 2.3.1 that after 60 days of seedling growth, leaf area, number of leaves per plant, length of stems and main roots, dry weight of stems and roots, and leaf content were measured (or observed). Why did the author not measure photosynthetic rate? Why is the above ground dry weight data not listed? If the author could supplement these data, the above results would be better explained.

8. Some citation method of the manuscript literature needs to be changed. For example, Lin 450: “[49] noted that”, line 465: “[50] and [52] demonstrated that”, etc.

Reviewer 4 Report

Comments and Suggestions for Authors

The work: ,,Unveiling the Synergistic Effects of Phosphorus Fertilization 2 and Organic Amendments in Mitigating Saline Water Irrigation and Dryland Stresses: Assessment of Red Pepper (Capsicum annuum L.) Growth, Productivity and Physio-Biochemical changes” has very long title. It’s too long and should be shorted. The organic amendments should be precise, and why phosphorus? Explain.

What do you mean like ,,organic amendments”? Some of Introduction should be cleared, and then consequently.

Material and methods – the subtitles 3.2.3 and 3.3.3. should be changed.

The methods should be described more exactly.

How do you measured soil salinity? Correct.

Figures and Tables, Results etc.: The units should be corrected.

Results and discussion: the subtitles should be changed.

Caption and footnotes of all figures and tables should be corrected. Explanations of abbreviations are needed additionally.

The Conclusions should be described strictly in relation to the research.

Comments on the Quality of English Language

Generally is correct.

Round 2

Reviewer 1 Report

Comments and Suggestions for Authors

After reviewing the article and explaining the authors' explanations, it may be published in Plants

Comments on the Quality of English Language

English is readable

Reviewer 2 Report

Comments and Suggestions for Authors

no comments